# Functional Divergence for N-Linked Glycosylation Sites in Equine Lutropin/Choriogonadotropin Receptors

**DOI:** 10.3390/cimb47080590

**Published:** 2025-07-25

**Authors:** Munkhzaya Byambaragchaa, Han-Ju Kang, Sei Hyen Park, Min Gyu Shin, Kyong-Mi Won, Myung-Hwa Kang, Kwan-Sik Min

**Affiliations:** 1Carbon-Neutral Resources Research Center, Anseong 17579, Republic of Korea; munkhzaya_b@yahoo.com; 2Genetic Engineering, Hankyong National University, Anseong 17579, Republic of Korea; gkswn111@hanmail.net; 3Graduate School of Animal BioScience, Hankyong National University, Anseong 17579, Republic of Korea; mrtree119@naver.com; 4Aquaculture Research Division, National Institute of Fisheries Science, Busan 46083, Republic of Korea; smg159@korea.kr (M.G.S.); kyongmiwon@korea.kr (K.-M.W.); 5Department of Food Science and Nutrition, Hoseo University, Asan 31499, Republic of Korea; mhkang@hoseo.edu; 6Division of Animal BioScience, School of Animal Life Convergence Sciences, Hankyong National University, Anseong 17579, Republic of Korea

**Keywords:** equine LH/CGR, N-linked glycosylation, cAMP responsive, cell surface loss of receptor, pERK1/2 activity

## Abstract

Equine lutropin hormone/choriogonadotropin receptor (LH/CGR) is a G protein-coupled receptor that binds to both luteinizing hormone and choriogonadotropin, with multiple potential N-linked glycosylation sites in the long extracellular domain region. The roles of these glycosylation sites in hormone binding have been widely studied; however, their relationships with cyclic adenosine monophosphate (cAMP) activation, loss of cell surface receptors, and phosphorylated extracellular signal-regulated kinases1/2 (pERK1/2) expression are unknown. We used site-directed mutagenesis with the substitution of Asn for Gln to alter the consensus sequences for N-linked glycosylation, and cAMP signaling was analyzed in the mutants. Specifically, the N174Q and N195Q mutants exhibited markedly reduced expression levels, reaching approximately 15.3% and 2.5%, respectively, of that observed for wild-type equine LH/CGR. Correspondingly, the cAMP EC_50_ values were decreased by 7.6-fold and 5.6-fold, respectively. Notably, the N195Q mutant displayed an almost complete loss of cAMP activity, even at high concentrations of recombinant eCG, suggesting a critical role for this glycosylation site in receptor function. Despite these alterations, Western blot analysis revealed that pERK1/2 phosphorylation peaked at 5 min following agonist stimulation across all mutants, indicating that the ERK1/2 signaling pathway remains functionally intact. This study demonstrates that the specific N-linked glycosylation site, N195, in equine LH/CGR is indispensable for cAMP activity but is normally processed in pERK1/2 signaling. Thus, we suggest that in equine LH/CGR, agonist treatment induces biased signaling, differentially activating cAMP signaling and the pERK1/2 pathway.

## 1. Introduction

Lutropin hormone/choriogonadotropin receptors (LH/CGRs), along with glycoprotein hormone receptors such as follicle-stimulating hormone receptors (FSHRs) and thyroid-stimulating hormone receptors (TSHRs), are G protein-coupled receptors (GPCRs). These receptors contain a large extracellular N-terminal domain responsible for ligand binding, a seven-transmembrane domain, and an intracellular domain containing Ser and Thr residues that undergo multiple phosphorylation modifications by G protein-coupled receptor kinases (GRKs) [1].

Post-translational modifications (PTMs), a fundamental phenomenon across all classes of life, play a crucial role in enhancing the biological functions of proteins and significantly increase their diversity [2]. One important class of proteins regulated by PTMs is GPCRs, which are expressed on the cell surface and play crucial roles in signal transduction and cellular communication. One of the most well-known types of glycosylation is N-linked glycosylation, which occurs at approximately 70% occupancy and plays crucial roles in protein folding, stability, and cellular signaling [3]. The well-established role of N-linked glycosylation in GPCRs is its contribution to receptor folding, trafficking, ligand responsiveness, and downstream signaling. As this modification occurs co-translationally in the endoplasmic reticulum, it plays a crucial role in ensuring proper protein folding and functional integrity [4].

Some studies have reported that the loss of N-glycosylation does not alter receptor function, beta-adrenergic receptors [5], human calcium receptors [6], or orphan GPR16 receptors [7], whereas others have shown that it impairs receptor binding and downstream signaling, human prostacyclin receptor [8], delta-opioid receptor [9], smoothened [10], mucin 16 [11], and orphan GPR 176 [12]. In recent studies, N-glycosylation mutagenesis in the extracellular domain attenuated cAMP signaling in the succinate receptor SUCNR1 [13] and β2-adrenoceptors [14]. Therefore, it can be understood that the function of N-glycosylation varies depending on the GPCRs.

Equine LH/CGR also has seven N-glycosylation sites in its extracellular domain [15]. Therefore, it is necessary to analyze the function of each glycosylation site. LH/CGR is mainly expressed in the ovaries and testes and plays a key role in gametocyte formation, follicle maturation, and ovulation [16]. In Leydig cells, it regulates testosterone production and supports spermatogenesis. It is also found in the theca cells, granulosa cells, and the corpus luteum [17]. These LH/CGRs activate adenylyl cyclase via Gs protein, increasing cAMP levels, and also trigger β-arrestin-mediated pathways [18,19]. β-arrestin links agonist-stimulated receptors, phosphorylated by GRKs, to downstream signaling. This is followed by the mitogen-activated protein kinase kinase kinase (MAP3K; c-Raf)—MAP2K (MEK1)—an extracellular signal-regulated kinases1/2 (ERK1/2) cascade, which ultimately leads to phosphorylated ERK1/2 (pERK1/2) [20].

In rat LH/CGR, N-linked oligosaccharides were not strictly required for hormone binding, as demonstrated by tunicamycin treatment [16] and each mutant [21]. However, the rat LH/CGR-Asn173 mutant is crucial for receptor trafficking, activity, and expression [22]. Similarly, rat FSHR does not require N-linked oligosaccharides for ligand binding [23], whereas human TSHR selectively depends on expression of the active receptor [24]. Therefore, in glycoprotein hormone receptors, N-linked glycosylation sites do not directly participate in receptor binding but are known to play a key role in selective cell surface expression and cAMP activation.

Different ligands can either trigger specific conformations of the same GPCR or activate distinct signaling pathways, such as G proteins or β-arrestins [25]. Thus, N-glycosylation in GPCRs plays a pivotal role in shaping biased signaling and functional selectivity. This study aimed to elucidate the functional role of N-linked glycosylation in receptor-mediated signal transduction, including changes in cAMP production and other downstream signaling events.

To date, no study has provided a comparative analysis of G protein signaling and pERK1/2 in relation to N-linked glycosylation sites. In this study, we investigate the impact of mutations at potential glycosylation sites (Asn50, Asn99, Asn174, Asn195, Asn291, Asn299, and Asn313) on receptor expression, receptor retention at the cell surface, and downstream signaling. Our findings demonstrate that N-glycosylation at Asn195 is essential for the cAMP response, receptor expression, and surface retention but not for pERK1/2 signaling.

## 2. Materials and Methods

### 2.1. Materials

Oligonucleotides used for mutagenesis were custom synthesized by Genotech (Daejeon, Republic of Korea). The polymerase chain reaction (PCR), DNA ligation, and restriction enzyme reagents were purchased from Takara Bio (Shiga, Japan). The pGEM-T easy subcloning vector was purchased from Promega (Madison, WI, USA). Fetal bovine serum (FBS) and Lipofectamine-2000 were obtained from Invitrogen (Carlsbad, CA, USA). The pCORON1000 SP VSV-G mammalian expression vector was purchased from Amersham Biosciences (Piscataway, NJ, USA). The CHO-K1 cells and HEK 293 cells were obtained from the Korean Cell Line Bank (KCLB, Seoul, Republic of Korea). Cell culture media including CHO-SFM II, Ham’s F-12, and OptiMEM were purchased from Gibco (Grand Island, NY, USA). The cAMP Dynamic 2 homogeneous time-resolved fluorescence (HTRF) assay kit was purchased from Cisbio (Codolet, France). rec-eCG, a ligand used in receptor activation assays, was produced in our laboratory, as described previously [26]. The QIAprep-Spin plasmid kit was obtained from Qiagen (Hilden, Germany). All other reagents were purchased from Sigma-Aldrich (St. Louis, MO, USA) and Wako Pure Chemical Industries (Osaka, Japan).

### 2.2. Construction of Glycosylation Mutants of Equine LH/CGR cDNA

Full-length equine LH/CG receptor cDNA was subcloned into the pGEM-T easy vector (Promega), as previously described [26]. Equine LH/CGR mutants were generated using an overlapping PCR. Seven potential N-linked glycosylation sites were predicted at the amino acid positions 50, 99, 174, 195, 291, 299, and 313. Each putative glycosylation site was modified by substituting an Asn (AAC) with a Gln (CAA). First, the mutant fragments upstream and downstream of the glycosylation site were amplified separately by PCR. These two PCR fragments were used as templates to ligate the upstream and downstream regions. The second PCR product, containing the XhoI and EcoRI restriction enzyme sites, was subcloned into the pGEM-T Easy vector and sequenced to verify PCR fidelity. Finally, equine LH/CGR mutants were subcloned into the expression vector pCORON1000 SP VSV-G (Amersham Biosciences). The signal sequence of the receptor was omitted, and the VSVG signal region from the expression vector was used. The plasmids were designated as equine LH/CGR-N50Q, -N99Q, -N174Q, -N195Q, -N291Q, -N299Q, and -N313Q. A representation of the mutation sites and various activation and inactivation sites in equine LH/CGR is shown in Figure 1.

### 2.3. Transient Transfection into CHO-K1 and HEK 293 Cells

CHO-K1 cells were cultured in Ham’s F-12 medium growth medium, supplemented with 2 mM glutamine, 50 μg/mL streptomycin, 50 U/mL penicillin, and 10% FBS. HEK 293 cells were cultured in Dulbecco’s modified Eagle’s medium (DMEM), supplemented with 10 mM HEPES, 50 μg/mL gentamicin, and 10% FBS. Transfection was performed in a 6-well plate when the cells reached approximately 80–90% confluence. Each equine LH/CGR mutant plasmid was mixed with the Lipofectamine reagent and incubated at room temperature for 15 min. Following two washes with cell culture medium, the mixture was slowly added dropwise to the cells. After 5 h, an equal volume of 20% FBS was added to obtain a final FBS concentration of 10%.

### 2.4. cAMP Analysis Using Homogeneous Time-Revolved Fluorescence

The analysis was performed according to the supplier’s cAMP dynamic 2 Homogeneous Time-Resolved Fluorescence assay protocol (Cisbio, Codolet, France) based on a method previously used in this laboratory [26]. Briefly, transfected cells were diluted in 0.5 mM IBMX to prevent cAMP degradation and seeded into 384-well plates at a density of 10,000 cells per well. Then, 5 μL of rec-eCG ligand was added and incubated for 30 min to stimulate the cells. cAMP-d2 and anti-cAMP-cryptate were added, and after 1 h, the fluorescence at 665 and 620 nm was measured using a TriStar^2^ LB942 microplate reader (BERTHOLD Technologies, Wildbad, Germany). Results were presented as Delta F% (665 nm/620 nm ratio) to indicate cAMP inhibition, and cAMP concentrations were quantified by comparing the measured values to a standard curve using GraphPad Prism v.6.0 (San Diego, CA, USA).

### 2.5. Cell Surface Expression and Agonist-Induced Cell Surface Loss

The expression and surface loss of equine LH/CGR were evaluated using ELISA, as previously described [27]. Briefly, cells were seeded into 96-well plates 24 h post-transfection. The following day, the cells were incubated with rec-eCG at concentrations of 100, 250, 500, and 1000 ng/mL for various time points (0, 5, 15, 30, 60, and 120 min). The cells were gently washed with DPBS, fixed with 4% paraformaldehyde for 5 min, and blocked for 30 min. The cells were incubated with rabbit anti-VSVG antibody (1:1000), followed by incubation with horseradish peroxidase (HRP)-conjugated anti-rabbit antibody (1:4000) for 1 h. In the cell surface loss experiment for equine LH/CGR wild-type (LH/CGR-WT) and its mutants, rec-eCG was applied at a concentration of 1000 μg/mL in a time-dependent manner and analyzed. Finally, the Substrate and Luminol/Enhancer were added to measure the luminescence signals. At time 0, the expression level was set to 100%, and receptor loss was defined as 0%.

### 2.6. pERK1/2 Activation by Western Blot Analysis

Western blotting was performed as previously described [26], using 5–10 μg of cell lysate protein. Briefly, cells were lysed using RIPA buffer (Sigma-Aldrich). Equal amounts of protein extract were loaded onto 12% SDS-PAGE gels and transferred to membranes. The membranes were incubated overnight at 4 °C with primary antibodies against pERK1/2 (1:2000) and total ERK1/2 (1:3000). The membrane was then incubated with an HRP-conjugated anti-rabbit secondary antibody for 1 h. The immune complexes were detected by chemiluminescence with SuperSignal^TM^ West Femto Maximum Sensitivity Substrate (Thermo Fisher Scientific, Waltham, MA, USA). The pERK1/2 immunoblots were quantified by densitometry using Image Lab v6.0 (Bio-Rad, Hercules, CA, USA). The antibodies used were p44/42 MAPK, anti-MAPK1/2, and goat anti-mouse-HRP, all from Cell Signaling Technology (Danvers, MA, USA).

### 2.7. Data Analysis

DNA sequence analysis was conducted using Multalin for multiple sequence alignment. cAMP concentration, derived from Delta F% raw data, along with EC_50_ values, and cAMP graphs, were analyzed using GraphPad Prism v.6.0. pERK1/2 graphs displaying percentage ratios and fold changes were generated using GraFit 5 software (Erithacus Software, Horley Surrey, UK). Statistical analysis was performed using one-way analysis of variance (ANOVA), followed by Tukey’s multiple comparison test using GraphPad Prism. Statistical significance between groups is indicated by * *p* < 0.05, ** *p* < 0.01.

## 3. Results

### 3.1. Mutation of Potential N-Linked Glycosylation Sites in Equine LH/CGR Modulates Receptor Expression

We generated mutant variants of equine LH/CGR by selectively altering putative N-linked glycosylation sites to investigate their impact on receptor expression and subsequent activation of intracellular signaling pathways. As shown in Figure 1, seven putative N-linked glycosylation sites were identified in the extracellular domain of equine LH/CGR. Specifically, an additional glycosylation site was uniquely present at position 50 in equine LH/CGR, which is absent in the LH/CGR of other mammalian species. The remaining six putative sites were conserved among mammals. The N201 site, located in exon 3 of the LH/CGR, and the other three sites (N291, N299, and N313), located in exon 10 of the LH/CGR, were identified by amino acid comparison analysis. Positions N174 and N195 are located in exons 6 and 7, respectively, and, based on current findings [22], have been reported to be the most critical glycosylation sites for signal transduction.

Thus, to elucidate the functional role of N-linked glycosylation in receptor-mediated signal transduction, including changes in cAMP production and other downstream signaling events, we modified the potential glycosylation sequence by substituting the Asn residue with Gln. The expression results shown in Figure 2 confirmed that five equine LH/CGRs (N99Q, N291Q, N299Q, and N313Q), including LH/CGR-WT, were efficiently expressed. However, three other mutants (N50Q, N174Q, and N195Q) exhibited considerably lower expression levels. The N50Q mutant was detected at approximately 38.3% relative to equine LH/CGR-WT. Notably, two mutants, N174Q and N195Q, showed a significant reduction in expression, with levels approximately 15.3 and 2.5%, respectively, that of equine LH/CGR-WT. Therefore, it is suggested that mutations the glycosylation sites at N174 and N195, located in exons 6 and 7, respectively, have a particularly significant impact on cell surface expression levels.

### 3.2. Biological Activities of Equine LH/CGR-WT and N-Glycosylation Mutants

The equine LH/CGR cAMP responses after rec-eCG agonist treatment are summarized in Figure 3 and Table 1. In equine LH/CGR-WT, the half-maximal effective concentration (EC_50_) and Rmax values were 23.7 ng/mL and 105.1 nM/10^4^ cells, respectively, showing a continuously increasing pattern in a dose-dependent manner.

The EC_50_ values for the N50Q and N291Q mutants were 17.7 ng/mL and 18.4 ng/mL, respectively, showing a slight decrease to 0.74- and 0.77-fold, respectively, compared to the equine LH/CGR-WT. However, the N291Q mutant exhibited only 0.72% of the Rmax of the equine LH/CGR. In the N99Q and N299Q mutants, the EC_50_ values were 24.1 ng/mL and 26.9 ng/mL, respectively, indicating a similarity to the equine LH/CGR-WT. However, the Rmax values of these two mutants were approximately 84 and 62%, respectively, that of the WT. The Rmax of the N313Q mutant, located in exon 10, was 90% of that of the WT. However, its EC_50_ value was approximately 1.5-fold higher, indicating a 50% reduction in sensitivity.

The N174Q mutant, located in exon 6, exhibited a significantly higher EC_50_ value of 179.4 ng/mL, which was 9.6-fold lower in sensitivity compared to the WT equine LH/CGR. The Rmax level was also measured at 65.6 nM/10^4^ cells, which was only 0.62-fold that of the WT. Therefore, this glycosylation site was found to have a significant impact on both EC_50_ and Rmax values when glycosylation was removed. Specifically, the cAMP response in the N195Q mutant was almost completely impaired, even at high concentrations of rec-eCG. The EC_50_ value and Rmax level for N195Q mutant were 135.6 ng/mL and 17.7 nM/10^4^ cells, respectively. These results showed that the EC_50_ value was 5.7-fold lower compared to the equine LH/CGR-WT, while the Rmax value was only 0.16-fold (Figure 4).

At specific glycosylation sites, the N174Q and N195Q mutants exhibited a significant reduction in both cAMP response and Rmax levels, suggesting that the substitution of Asn with Gln at these sites has a considerable impact on cAMP signaling. Thus, we determined that although not all N-linked glycosylation sites are required for cAMP responsiveness, the specific glycosylation sites at Asn174 and Asn195 play crucial roles in cAMP signal transduction. Therefore, the N174Q and N195Q mutants exhibited only 15.3% and 2.5%, respectively, that of the WT receptor expression on the cell surface. Additionally, their cAMP levels were reduced by 9.6- and 5.6-fold, respectively, indicating that these two glycosylation sites play a critical role in receptor function.

The absence of glycosylation at these sites, combined with their substitution with glutamine, is believed to induce conformational changes during PTMs. Thus, the potential glycosylation sites at Asn174 and Asn195, located in exons 6 and 7 of the equine LH/CGR, respectively, may serve as key models for elucidating the structural mechanisms of equine LH/CGR in receptor-mediated signaling complexes.

### 3.3. Downregulation of Cell Surface Receptors Following Agonist Stimulation

Loss of receptor surface expression was quantitatively analyzed using ELISA after treatment with rec-eCG, following a previously described protocol [27]. Cells expressing equine LH/CGR-WT were stimulated with the agonist for 120 min to evaluate the dose-dependent reduction in receptor expression on the cell surface. This allowed for a comprehensive analysis of receptor downregulation in response to varying agonist concentrations (Figure 5). Therefore, since 1 µg/mL was found to be the most effective concentration for observing receptor loss, further experiments were conducted in a time-dependent manner using this concentration.

In cells expressing the equine LH/CG-WT, receptor levels decreased by approximately 84% within the first 5 min, followed by a gradual reduction in receptor loss, ranging from 67 to 75% until 120 min. In the two mutants, the equine LH/CGR-N50Q and -N174Q, the surface receptor levels decreased to 76 and 69%, respectively, within the first 5 min. Thereafter, the levels further declined to 63 and 56% within 15 min and remained consistently between 51 and 68% until 120 min. These two mutants exhibited a more rapid loss of surface receptors than the WT. In particular, the N174 mutant showed only 15.3% of the surface expression level compared to the control and had a cAMP EC_50_ value that was 9.6-fold lower. Despite these characteristics, surface receptor loss occurs more rapidly. This phenomenon was likely due to the extremely low surface expression of the receptor (Figure 6).

Receptor loss in cells expressing the N99Q mutant followed a pattern similar to that observed in WT cells, with comparable kinetics, and the extent of receptor reduction over the experimental time course. The three mutants (N291Q, N299Q, and N313Q) located in exon 10 showed minimal receptor surface loss, with receptor levels remaining at 92–94% within the first 5 min. Thereafter, the receptor levels gradually decreased, reaching 75–79% after 30 min. From 60 min to 120 min, the reduction continued at a slow rate, resulting in levels ranging from 64–70%.

Specifically, the expression of the surface receptor in the N195Q mutant was approximately 54% within 5 min, but its expression level subsequently increased to 76% at 15 min. At 30 min, the levels sharply increased to 120% and thereafter remained nearly constant with minimal fluctuations. This mutant exhibited approximately 2.5% of the expression level relative to the equine LH/CGR-WT and showed the lowest cell surface loss. This mutant exhibited very low expression levels, resulting in a decrease in surface receptors only during the initial phase of agonist treatment. However, between 15 and 30 min, the receptors were presumed to have been internalized and rapidly recycled back to the cell surface.

When considering only receptor loss, the reduction in cell surface receptors at 30 min was 34% in equine LH/CGR-WT cells, whereas receptor loss was 0% in ligand-pretreated cells (Figure 7). The N50Q and N174Q mutants exhibited a slight increase in receptor loss (44–45%) compared to the WT. Cells expressing the N291Q, N299Q, and N313Q mutants showed an approximately 25–26% reduction in receptor loss at 30 min. In the N174Q mutant, despite its low expression, the surface receptor loss reached approximately 44%, whereas no receptor loss was observed in cells expressing the N195Q mutant at 30 min.

The rate of formation of equine CG-LH/CGR complexes induced by the N-glycosylation mutants is presented in Table 2. The formation rate was rapid in both the WT and the N50Q, N99Q, and N174Q mutants, with t_1_/_2_ values ranging from 3.3 min to 5.6 min. The other three mutations (N291Q, N299Q, and N313Q) located in exon 10 exhibited a two-fold delay compared to the equine LH/CGR-WT. Exceptionally, the N195Q mutant showed a significantly prolonged delay, with a t_1_/_2_ of approximately 31.4 min, indicating a 5.6-fold slower rate than the equine LH/CGR-WT (Table 2).

The equine LH/CGR-N195Q mutant exhibited the slowest receptor loss, which was attributed to its extremely low expression. Additionally, nearly impaired cAMP signaling is suspected to contribute to receptor loss. Therefore, in the N195Q mutant, the substitution of Asn with Gln at the glycosylation site appears to significantly affect receptor expression and downregulation owing to alterations in PTMs.

### 3.4. pERK1/2 Activation in HEK293 Cells

Next, we analyzed pERK1/2 levels in HEK-293 cells transiently transfected with the equine LH/CGR mutants. We also conducted a time-dependent analysis of pERK1/2 activity by Western blotting. Rapid pERK1/2 activation was detected in equine LH/CGR-WT cells at 5 min and gradually decreased after 30 min. The pERK1/2 activity was detected at approximately 40% of its maximum level within 30 min. In the N99Q, N174Q, N195Q, and N313Q mutants, the overall time-dependent pattern of pERK1/2 activity was slightly lower than that in the WT.

All the four mutants peaked at 5 min, followed by a gradual decline over time. Specifically, compared to the WT, the N99Q and N313Q mutants were observed at 53 and 47%, respectively, at the 5 min mark (Figure 8A,B). However, the N174Q and N195Q mutants exhibited pERK1/2 activity at 90 and 75% at 5 min. Notably, in the N174Q and N195Q mutants, despite the absence of cell surface expression, cAMP activation, and cell surface loss, the pERK1/2 signaling pathway was initially intact. After 15 min, pERK1/2 activity significantly decreased to approximately 31–40% relative to its level at the 5 min mark in the equine LH/CGR-WT.

In the present study, we demonstrated that the two signaling pathways are distinctly differentiated based on their responses to cAMP levels and pERK1/2 activation. Therefore, we propose that in the N174Q and N195Q equine LH/CGR mutants, which involve N-linked glycosylation sites, the cAMP and pERK1/2 pathways exhibit pleiotropic signal transduction in response to agonist-induced stimulation with recombinant eCG (Figure 9). In summary, among the N-linked glycosylation sites of the equine LH/CGR, the N174Q and N195Q mutants exhibited distinctly biased signaling, specifically in terms of the cAMP response and pERK1/2 activity. This suggests that these glycosylation sites play a crucial role in modulating differential signal transduction pathways within the receptor.

## 4. Discussion

The current study aimed to elucidate the functions of glycosylation sites located in the extracellular domain of the equine LH/CGR. We specifically examined their roles in receptor expression, cell surface receptor loss, cAMP signaling, and pERK1/2 activity. LH/CGR consists of an extracellular domain with leucine-rich repeats followed by a rhodopsin-type Class A seven-transmembrane helix domain. There are potential N-linked glycosylation sites in the extracellular domain, and most mammals have six highly conserved sites. However, in the equine LH/CGR, an additional site is present at amino acid position 50 of exon 1. In the mammalian LH/CGR, well-conserved potential N-linked glycosylation sites are present in exons 3, 6, and 7, whereas exon 10 contains three such sites.

In the present study, the N174Q and N195Q mutants exhibited significantly lower expression and the lowest cAMP response. These results are consistent with those of previous studies; rat LH/CGR-Asn173 corresponds to the same position as residue 195 in the equine LH/CGR to exclude the signal sequence, resulting in a decreased ability of the receptor to be expressed on the plasma membrane [22]. Additionally, previous studies on rat LH/CGR-Asn173 have shown that it is essential for hormone binding [28]. Deglycosylation of the mature receptor with neuraminidase preserved ligand-binding capacity in Leydig tumor cells and rat granulosa cells but failed to bind when treated with tunicamycin B2 [16]. Rat LH/CGR-N152 and -N173 carry functional carbohydrate chains when expressed in insect cells. Tunicamycin treatment of the WT and these two mutants resulted in no hormone-binding activity [29]. They also reported that three mutations (Asn269, Asn277, and Asn291, located in exon 10) were not glycosylated and did not contribute to hormone binding. Although ligand-binding experiments were not conducted in this study, the carbohydrate chains at the equine LH/CGR-Asn174 and -Asn195 sites, located in exons 6 and 7, were essential for the proper expression of a functional receptor on the cell surface. These glycosylation sites play a crucial role in the intramolecular folding of nascent receptors and their absence results in an impaired cAMP response.

The N50Q mutant, located in exon 3—which is not present in other mammalian receptors—exhibited distinct outcomes in both expression levels and cAMP production. Although the cell surface expression of the N50Q mutant was significantly reduced by approximately 38%, its cAMP production was paradoxically increased. This unique observation, specific to this mutant, suggests that there may not be a direct correlation between cell surface expression and cAMP production. One possible explanation is that this region lies at the very beginning of the exon, which may have limited impact on post-translational modifications (PTMs), while promoting a folding structure that favors PKA activation.

The importance of the Asn174 and Asn195 glycosylation sites, as demonstrated in the present study, is specifically related to receptor expression and the cAMP response. Therefore, we examined whether these results were correlated with the loss of receptors on the cell surface. The equine LH/CGR-N195Q mutant specifically delayed receptor loss from the cell surface, whereas the equine LH/CGR-N174Q mutant exhibited an even faster loss compared to the WT (Table 2). Although these mutants exhibited similarly low expression levels and cAMP signaling, they showed different results regarding receptor loss on the cell surface. This suggests that these mutants are likely directly involved in the folding process during PTMs. These findings indicate that not all N-linked glycosylation sites of the equine LH/CGR are essential; instead, distinct glycosylation sites selectively contribute to specific receptor functions, such as proper trafficking to the cell surface, regulation of receptor internalization, and modulation of PKA-mediated physiological responses. Recent structural studies on the human LH/CGR have proposed a push-and-pull mechanism for its activation in the LH/CGR produced in insect cells. The highly conserved 10-residue fragment (amino acids 350–359) between the extracellular and transmembrane domains plays a crucial role in inducing conformational changes for G protein coupling [30]. The small allosteric molecule 21f activated agonist-mediated FSHR based on the structure of human FSHR, highlighting a conserved mechanism of hormone-induced receptor activation [31]. This finding indicates that similar hinge loop–hormone interactions are observed in both the CG-LH/CGR-Gs and TSH-TSHR-Gs complexes [30,32]. They also suggested that the extracellular domain, when in its inactive state, is tilted toward the membrane and positioned almost perpendicular to it. These inactive structures have been reported in class A GPCRs, including rhodopsin [33] and the β2-adrenergic receptor [34], particularly at transmembrane helix 6. Naturally occurring inactivating mutants are predominantly found in the transmembrane domains of glycoprotein hormone receptors. However, our results suggest that the two N-linked glycosylation sites located in exons 6 and 7 induce a conformational change when Asn is substituted with Gln, ultimately leading to an inactive structure.

In previous studies on FSHR and TSHR, the rat FSHR-N174Q mutant, located at the same exon 7 position as Asn195 in the equine LH/CGR, was expressed on the cell surface and bound to FSH with normal affinity [23]. Thus, the N-linked glycosylation of human FSHR is not directly required for hormone binding. However, the eel FSHR-N191Q mutant exhibits completely impaired signaling and only 9% expression compared to the WT [27]. Human TSHR-N77Q and N113Q mutants showed markedly decreased binding affinities. In contrast, the N177Q and N198Q mutants did not significantly affect receptor binding affinity but led to an increase in the intracellular cAMP response [24]. Therefore, among the glycoprotein hormone receptors, the highly conserved N-linked glycosylation site at Asn195, located in exon 7, plays a crucial role in signal transduction after PTMs. These results, along with the data for the equine LH/CGR-N195Q, suggest that the N-glycosylation site is critical for receptor expression, receptor surface loss, and particularly, the cAMP response. The N195Q mutant shows a distinctly different result compared to other glycosylation site mutants. This unique outcome may be attributed to the significantly reduced receptor expression observed with the N195Q mutation, which is likely closely related to improper receptor folding. As a result, it can be concluded that there is minimal loss at the cell surface in this mutant. Although the receptor binding analysis was not performed in this study, the N195Q mutant was found to have the greatest impact on cAMP activity. Therefore, it is suggested that the glycosylation sites play important roles in receptor stability, turnover, and interactions with other proteins.

In many other GPCRs, N-glycosylation mutations reduce receptor surface expression, as reported for human protease-activated receptor-1 [35] and PAR1 [36]. However, other receptors are normally expressed, such as the muscarine M2 receptor [37] and class A orphan receptor GPR61 [7]. Recent studies on N-linked deglycosylation in class C GPCR, specifically GPRC6A, have shown that one of the seven N-glycan sites plays a crucial role in regulating receptor surface expression, whereas the others contribute to various aspects of receptor function, such as signaling efficiency [38]. Therefore, these N-linked glycosylation sites appear to play distinct roles in receptor expression and signaling, with specific functions depending on the glycosylation site. Moreover, studies have shown that glycosylation functions vary among GPCRs rather than being uniformly preserved across all receptors.

ERK1/2 activation is widely known to be driven by β-arrestin binding to the phosphorylated intracellular domain of GPCRs through GRK-mediated signaling [39,40]. As a result, biased agonists present a unique opportunity to regulate signaling pathways by selectively activating either G protein- or β-arrestin-mediated signaling [41,42,43]. In the present study, pERK1/2 activity was detected in all the mutants, peaking at 5 min. Specifically, the two mutant cell lines expressing N174Q and N195Q showed a significant increase within 5 min, similar to the WT. However, these two mutants showed a significantly greater reduction in pERK1/2 signaling than the WT at 15 min and 30 min, respectively, suggesting a stronger effect on late-phase signaling than on the initial response.

These findings are consistent with previous reports on hCG- and FSH-induced activation, which similarly exhibited peak activity within approximately 5–6 min after receptor stimulation [18,27]. These two mutants exhibited very low expression and cAMP activity, yet pERK1/2 signaling proceeded normally. This result is consistent with previous findings on active and inactive receptor states [44]. Therefore, it has been confirmed that pERK1/2 activation occurs through other pathways than the LH/CGR-mediated PKA signaling pathway, such as β-arrestin, indicating biased signaling. Thus, N-linked glycosylation sites on glycoprotein hormone receptors are not essential for pERK1/2 activation. This mutant serves as an ideal model for investigating biased signaling because it distinctly exhibits differential activation of downstream pathways. While the cAMP response and overall receptor expression were significantly reduced, the ability to activate pERK1/2 signaling remained intact. This unique signaling profile highlights the potential for exploring the molecular mechanisms underlying biased agonism and receptor functionality in GPCR-mediated pathways. Recent studies on biased antagonist signaling and agonism have reported findings related to the glycoprotein hormone receptors for LH/CG and FSH [45,46,47,48]. These results align with previous findings demonstrating the selective activation of β-arrestin-dependent signaling without engaging in G protein coupling [49,50]. The inactive FSHR-A189V and -M512I mutants, which exhibit impaired G protein signaling, instead activate β-arrestin-dependent pERK1/2 signaling, demonstrating biased signaling [51,52].

Consequently, the N-linked glycosylation sites of the equine LH/CGR may play a selective role in modulating PKA signaling and pERK1/2 activation through biased signaling mechanisms. While the specific N-linked glycosylation site (Asn195) is crucial for cell surface expression, PKA signaling, and receptor surface loss, its effect on pERK1/2 signaling remains minimal in the early phase but gradually decreases in the later phase. In conclusion, both cAMP and pERK1/2 signaling are modulated in a site-specific manner, depending on the specific N-linked glycosylation site. Among the various PTMs, glycosylation sites are considered key determinants that play crucial roles in the structural integrity and functional regulation of proteins.

## 5. Conclusions

Taken together, this study demonstrated that the N195Q mutants had the lowest expression on the cell surface, followed by a basal cAMP response, which was not completely impaired. However, pERK1/2 activity proceeded normally in a time-dependent manner despite a slight decrease in the later phase. Although this study did not directly examine structural changes at the molecular level, it is presumed that the absence of glycosylation at this specific site, combined with the substitution of asparagine with glutamine, may have led to conformational alterations in the receptor structure following post-translational modifications (PTMs). These structural shifts could, in turn, influence receptor folding, trafficking, and overall functionality. In this context, the potential N195 glycosylation site, located within exon 7 of the LH/CGR, may represent a valuable model system for exploring the mechanistic roles of site-specific glycosylation in modulating receptor-mediated signaling. Specifically, it could provide important insights into how the disruption of glycosylation can impair cAMP signaling efficiency and alter downstream regulatory pathways involved in the GPCR function. This finding indicates that pERK1/2 activity operates independently of the G protein-mediated cAMP response. Accordingly, we propose that the pERK1/2 signaling pathway can be activated through an alternative mechanism, independent of G protein signaling. These findings shed light on the regulatory significance of N-linked glycosylation sites in the equine LH/CGR function and provide a deeper understanding of GPCR signal transduction mechanisms.

## Figures and Tables

**Figure 1 cimb-47-00590-f001:**
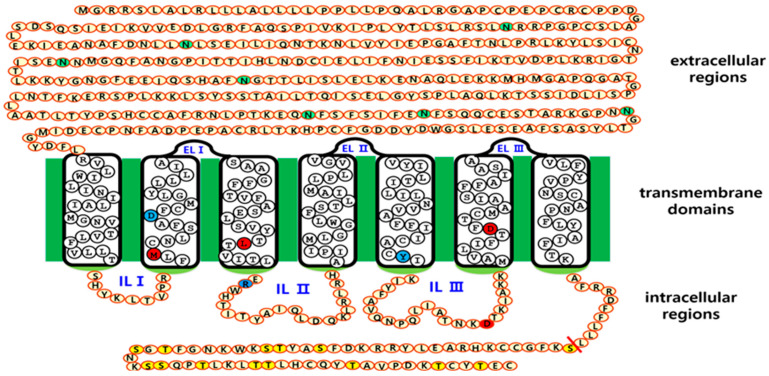
Schematic representation of the structure of equine LH/CGR and its N-linked glycosylation sites. Equine LH/CGR contains seven potential N-linked glycosylation sites in its extracellular domain. Among these, six sites (Asn99, Asn174, Asn199, Asn291, Asn299, and Asn313) are highly conserved among mammals. Additionally, Asn50 is specifically located in the anterior region of the equine LH/CGR N-terminus. Notably, Asn174 and Asn199 are located in exons 6 and exon 7, respectively. Residues Asn291, Asn299, and Asn313 are commonly located in exon 10. Several constitutively activating (M398T, I457R, D564G, and D578Y) and inactivating (D405N, R464H, and R546F) mutations have been identified. These mutations are marked with red and blue circles, indicating their activation and inactivation effects, respectively. In the intracellular domain, specific Ser and Thr residues in the C-terminal region are phosphorylated by G-protein-coupled receptor kinases (GRKs). Subsequently, β-arrestin binds to the phosphorylated region, leading to downstream signal transduction. Given this regulatory mechanism, signal transduction was analyzed in relation to glycosylation site variants. The green circles indicate N-linked glycosylation sites in the extracellular region. Red circles represent the four sites associated with constitutive activating mutants. Blue circles denote inactivating sites among mammalian LH/CGRs. Yellow circles show phosphorylation sites in the intracellular region.

**Figure 2 cimb-47-00590-f002:**
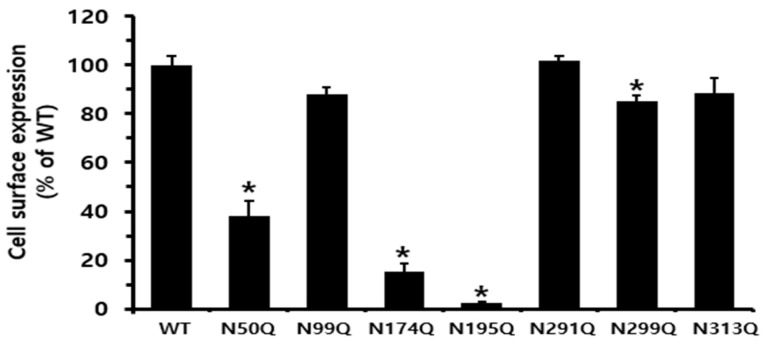
Cell surface expression of equine LH/CGR at N-linked glycosylation sites in HEK 293 cells transiently transfected. Expression levels were determined using enzyme-linked immunosorbent assay (ELISA). Cell surface expression was measured in HEK 293 cells transiently transfected using an anti-VSVG tag. Values are presented as mean ± standard error of mean (SEM) from three independent experiments and normalized to equine LH/CGR-WT, which is set at 100%. Statistically significant differences were determined using one-way ANOVA, followed by Tukey’s comparison test (* *p* < 0.01).

**Figure 3 cimb-47-00590-f003:**
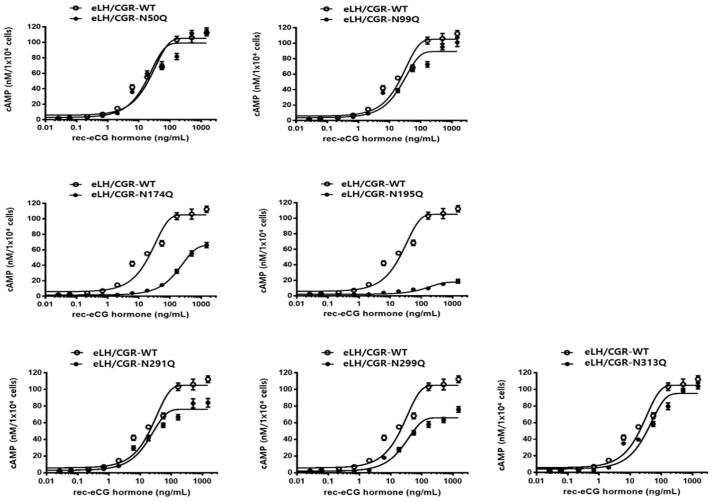
Total cAMP levels in CHO-K1 cells transfected with equine LH/CGR N-linked glycosylation site mutants. After 48 h of transfection in a 6-well plate, 10,000 cells were seeded in a 384-well plate. Cells were stimulated with rec-eCG in a dose-dependent manner for 30 min and cAMP levels were assessed using a homogeneous time-resolved fluorescence (HTRF) assay. To determine Delta F%, the ratio of values measured at 665 and 620 nm was calculated, providing a quantitative assessment of the fluorescence signal intensity. The obtained data were analyzed against a cAMP standard curve to quantify cAMP production using GraphPad Prism software for precise calculations. Data represent mean ± standard error of the mean (SEM) of three independent experiments. Mean values were fitted to a single-phase exponential decay curve for analysis.

**Figure 4 cimb-47-00590-f004:**
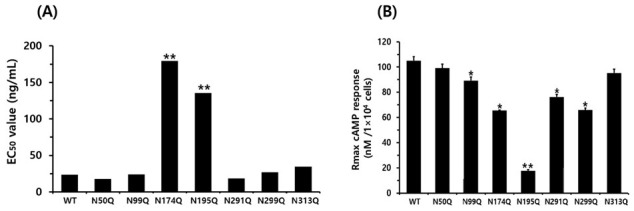
EC_50_ value and Rmax level of total cAMP for equine LH/CGR N-linked glycosylation mutants. cAMP EC50 (**A**) and Rmax (**B**) for each mutant, as presented in Figure 3 and Table 1, are visually represented as bar graphs to facilitate data comparison and interpretation. Asn174Q and Asn195Q mutants exhibited a significant decrease in both EC50 and Rmax, indicating a substantial impact on receptor function and cAMP signaling efficiency. Statistically significant differences were determined using one-way ANOVA, followed by Tukey’s comparison test (* *p* < 0.05, ** *p* < 0.01).

**Figure 5 cimb-47-00590-f005:**
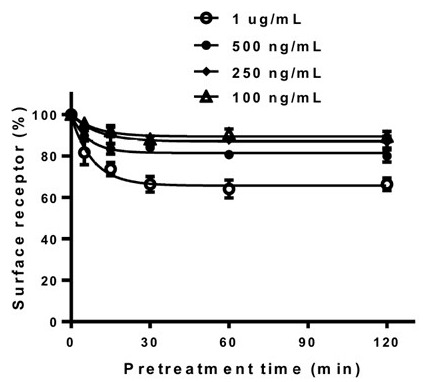
Dose-dependent regulation of cell surface loss in the equine LH/CGR-WT. After transfection, rec-eCG was added in a dose-dependent manner for up to 120 min at concentrations of 100 ng/mL, 250 ng/mL, 500 ng/mL, and 1 µg/mL. Cell surface receptor loss was analyzed using an anti-VSVG-tag antibody as described in the Section 2.

**Figure 6 cimb-47-00590-f006:**
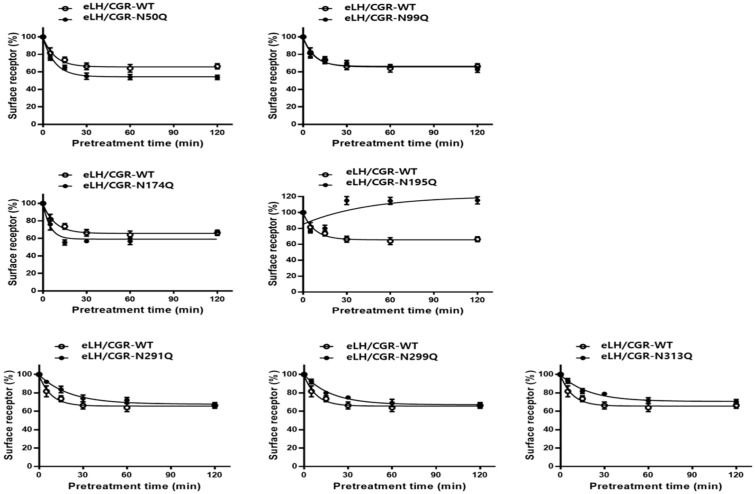
Time-dependent cell surface loss in the equine LH/CGR-WT and N-linked glycosylation mutants. The cell surface expression level at 0-time was set to 100% for comparison. Surface loss of receptor was measured for up to 120 min. Mean data were fitted to a single-phase exponential decay equation. Blank circles show the same curves for the equine LH/CGR-WT.

**Figure 7 cimb-47-00590-f007:**
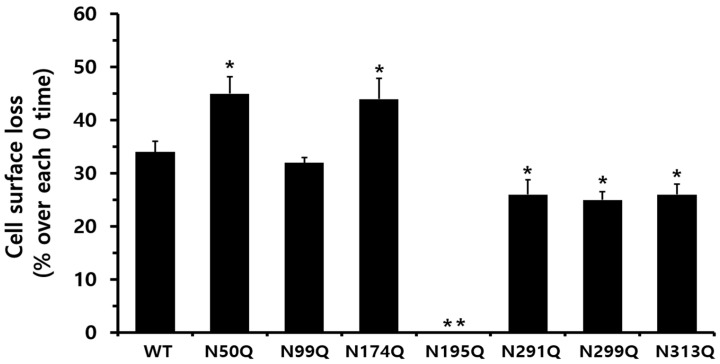
Comparison of cell surface loss ratios between the equine LH/CGR-WT and N-linked glycosylation mutants. Surface loss of a receptor was expressed as a percentage; without agonist treatment, it was considered as 0% loss. The decrease in surface loss is presented as a percentage at 30 min after agonist treatment. Statistically significant differences were determined using one-way ANOVA, followed by Tukey’s comparison test (* *p* < 0.05, ** *p* < 0.01).

**Figure 8 cimb-47-00590-f008:**
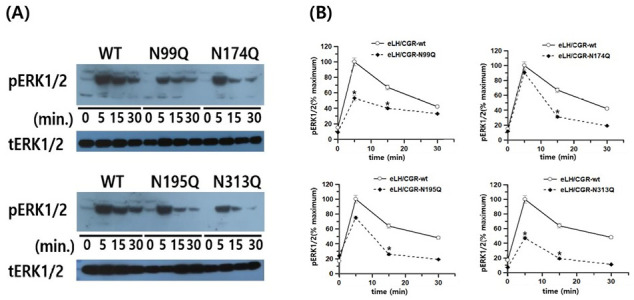
pERK1/2 activation in HEK 293 cells transfected with the equine LH/CGR-WT and N-linked mutants after stimulation with the rec-eCG agonist. After transfection of HEK 293 cells with each plasmid, cells were incubated under standard conditions to allow for proper expression. Following transfection, cells were serum-starved for approximately 6 h to minimize background signaling before stimulation with the rec-eCG agonist for designated time periods. The whole cell lysates were subsequently extracted to assess protein expression and signaling activity. The lysates were then analyzed to measure phosphorylated ERK1/2 (pERK1/2) and total ERK levels, allowing for evaluation of activation. (**A**) Western blotting results for pERK1/2 and total ERK levels. (**B**) Quantification of pERK1/2 levels. Data, normalized to total ERK levels, are presented as a percentage of the maximal response, with peak response for the equine LH/CGR-WT at 5 min set to 100%. Densitometric analysis was conducted to accurately quantify the relative intensity of pERK1/2 and total ERK bands, allowing for a precise assessment of protein phosphorylation and expression levels. Statistically significant differences were determined using one-way ANOVA, followed by Tukey’s comparison test (* *p* < 0.05).

**Figure 9 cimb-47-00590-f009:**
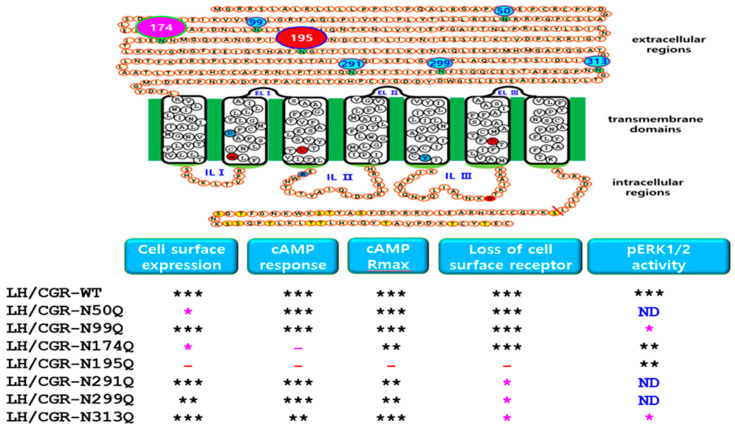
Functional summary of cell surface expression and downstream signaling in the equine LH/CGR-WT and N-linked glycosylation mutants. Results related to seven potential glycosylation sites are summarized as follows: Cell surface expression was significantly decreased in Asn50 and Asn174 mutant cells; however, the Asn195Q mutant showed almost no expression. cAMP responses in Asn174Q and Asn195Q mutants were 7.6- and 5.6-fold lower, respectively, than in the WT. No significant surface loss of receptor was observed in Asn195Q mutants, indicating that mutation may impair receptor trafficking or internalization. Three mutants (Asn291Q, Asn299Q, and Asn313Q) located in exon 10 exhibited a delay of more than two-fold compared to the WT. pERK1/2 activity in mutants was generally processed normally, similar to that in the WT at 5 min mark. However, the N99Q mutant exhibited a slightly reduced activity, indicating a minor deviation from the typical signaling response. In conclusion, potential N-linked glycosylation sites selectively influenced cell surface expression, receptor surface loss, and cAMP signaling. Given that pERK1/2 signaling appears to function normally in these mutants, this suggests that eLH/CGR exhibits biased signaling. Therefore, these mutants could serve as important models for studying biased signaling mechanisms. ND: Not used in the pERK1/2 experiment. The circle numbers indicate the six N-linked glycosylation sites in the extracellular region. The red circle numbered 195 is particularly important for expression levels and cAMP activity. The pink circle numbered 174 shows a significant reduction in cell surface expression and cAMP response. * indicates a statistically significant difference compared to the control (*p* < 0.01). ** indicates a statistically significant difference compared to the control (*p* < 0.05). *** indicates no statistically significant difference.

**Table 1 cimb-47-00590-t001:** Bioactivity of eLH/CG receptors in cells expressing receptor mutants.

eLH/CG Receptors	cAMP Responses		
	Basal *^a^* (nM/10^4^ Cells)	EC_50_ Value (ng/mL)	Rmax *^b^* (nM/10^4^ Cells)
eLH/CGR-WT	4.8 ± 1.9	23.7 (1.0-fold) (18.7 to 32.2) *^c^*	105.1 ± 3.0 (1-fold)
eLH/CGR-N50Q	2.9 ± 1.8	17.7 (0.7-fold) (13.5 to 25.9)	99.1 ± 3.3 (0.94-fold)
eLH/CGR-N99Q	3.8 ± 1.9	24.1 (1.0-fold) (18.6 to 34.0)	89.2 ± 2.9 (0.84-fold)
eLH/CGR-N174Q	1.7 ± 0.3	179.4 (7.6-fold) (165.6 to 195.6)	65.6 ± 0.3 (0.62-fold)
eLH/CGR-N195Q	2.1 ± 0.3	135.6 (5.7-fold) (100.9 to 207.0)	17.7 ± 0.8 (0.16-fold)
eLH/CGR-N291Q	2.9 ± 1.5	18.4 (0.7-fold) (14.5 to 24.9)	76.2 ± 2.1 (0.72-fold)
eLH/CGR-N299Q	1.8 ± 0.9	26.9 (1.1-fold) (22.2 to 49.2)	65.9 ± 1.6 (0.62-fold)
eLH/CGR-N313Q	3.3 ± 1.8	34.7 (1.5-fold) (27.2 to 47.9)	95.2 ± 3.0 (0.90-fold)

Values are the means ± SEM of three experiments. The half-maximal effective concentration (EC_50_) values were determined from the concentration–response curves from in vitro bioassays. *^a^* Basal cAMP level average without agonist treatment. *^b^* Rmax average cAMP level/10^4^ cells. *^c^* Geometric mean (95% confidence limit).

**Table 2 cimb-47-00590-t002:** Rates of cell surface loss of receptors in transiently transfected cell lines expressing the wild-type eLH/CGR and mutants thereof.

Ligand Treatment	eLH/CGR Cell Lines	t_1/2_ (min)	Plateau (% of Control)
rec-eCG	eLH/CGR-WT	5.6 ± 0.1	65.6 ± 1.4
	eLH/CGR-N50Q	5.6 ± 0.1	54.4 ± 1.1
	eLH/CGR-N99Q	5.2 ± 0.2	66.4 ± 1.3
	eLH/CGR-N174Q	3.3 ± 0.1	59.1 ± 1.9
	eLH/CGR-N195Q	31.4 ± 0.1	120.9 ± 12.5
	eLH/CGR-N291Q	14.3 ± 0.1	67.8 ± 1.4
	eLH/CGR-N299Q	12.8 ± 0.1	67.2 ± 1.1
	eLH/CGR-N313Q	13.3 ± 0.1	70.7 ± 1.3

Data were fitted to one-phase exponential decay curves to obtain values of t_1/2_ and plateau (i.e., maximum reduction). The data were from three individual experiments.

## Data Availability

All relevant data are contained within the article.

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
