# Peer review of "Functional Divergence for N-Linked Glycosylation Sites in Equine Lutropin/Choriogonadotropin Receptors"

_cimb, 2025, doi:10.3390/cimb47080590_

Round 1
Reviewer 1 Report
Comments and Suggestions for Authors
In this paper, the authors constructed LH/CGR mutants with a substitution of Asn to Gln to alter the potential N-linked glycosylation. They analyzed their effects on cell surface expression, cAMP activation, and pERK1/2 expression, and concluded that the N195Q mutant had the lowest expression on the cell surface with a basal cAMP response and normal pERK1/2 activity. However, there some some major issues that need to be addressed to support their conclusion.
1. The whole expression level of both wild-type and LH/CGR N-to-Q mutants should be examined to determine whether the N-to-Q mutant affects the expression level or stability of the expressed protein. This assay can be combined with MG-132 treatment or other protein degradation inhibitors
2. To test whether it is the expression level of LH/CGR or the site mutant that determines the cell surface expression and downstream cAMP response, siRNA-mediated downregulation of wild-type LH/CGR should be included in all the assays tested in this paper. Rescue experiment should be also performed.
Author Response
Reviewer 1
In this paper, the authors constructed LH/CGR mutants with a substitution of Asn to Gln to alter the potential N-linked glycosylation. They analyzed their effects on cell surface expression, cAMP activation, and pERK1/2 expression, and concluded that the N195Q mutant had the lowest expression on the cell surface with a basal cAMP response and normal pERK1/2 activity. However, there are some major issues that need to be addressed to support their conclusion.
- The whole expression level of both wild-type and LH/CGR N-to-Q mutants should be examined to determine whether the N-to-Q mutant affects the expression level or stability of the expressed protein. This assay can be combined with MG-132 treatment or other protein degradation inhibitors.
→ In the Discussion section, it was noted that the markedly low expression level observed in the N195Q mutant may be attributed to reduced receptor stability. As the reviewer correctly pointed out, additional experiments involving MG-132 treatment and other proteasome inhibition assays would be essential to further support this finding. However, due to current limitations, it is not feasible to conduct these experiments at this time. We kindly ask for your understanding regarding this matter.
- To test whether it is the expression level of LH/CGR or the site mutant that determines the cell surface expression and downstream cAMP response, siRNA-mediated downregulation of wild-type LH/CGR should be included in all the assays tested in this paper. Rescue experiment should be also performed.
→ We are currently preparing to conduct siRNA-mediated downregulation experiments, along with corresponding rescue experiments to restore gene function. In addition, we are in the process of establishing β-arrestin 1 and 2 as well as GRK2–6 knockout cell lines. These resources will enable us to systematically investigate and clarify the correlation between cAMP and pERK1/2 signaling in future studies.
Reviewer 2 Report
Comments and Suggestions for Authors
In this manuscript, Byambaragchaa and coworkers systematically determine the functional roles of the seven N-glycosylation sites in equine lutropin hormone/choriogonadotropin receptor. The study is well-written, logical and would be of interest to the readership of this journal. I recommend acceptance after the following comments have been addressed.
- Can the authors cite the literature showing the LH/CGR protein has seven N-glycosylations?
- The authors should mount the study better by reframing the abstract. Instead of detailing the results, they should specify the broader implications of this study and how this knowledge will benefit the scientific community.
- The authors dedicate significant portion of the manuscript to a single experiment measuring cAMP levels in presence of the LH/CGR mutants. They present the data as individual graphs, in a table and then in a bar plot (Figure 3, 4 and Table 1). I suggest that the authors only stick to the bar plot and move the graphs and table to the SI.
- The same point also applies to Figure 6, 7 and Table 2. In this case, Table 2 seems to represent the most relevant information. Please move the figures to SI.
- Can the authors discuss why the trend in N195Q is completely different compared to the other mutants?
- The major concern in this work is that the authors do not verify the presence of the N-glycosylation PTM, and whether it is affected by introducing the mutants. The authors should carry out an orthogonal measurement, such as mass spectrometry to determine the extent of N-glycosylation. Alternatively, they should cite literature showing that the side-directed mutagenesis does not affect N-glycosylation at the other sites.
- Line 361 has the first and only mention of N201Q. I assume this is a typo and should actually be N195Q.
Author Response
Reviewer 2
In this manuscript, Byambaragchaa and coworkers systematically determine the functional roles of the seven N-glycosylation sites in equine lutropin hormone/choriogonadotropin receptor. The study is well-written, logical and would be of interest to the readership of this journal. I recommend acceptance after the following comments have been addressed.
- Can the authors cite the literature showing the LH/CGR protein has seven N-glycosylations?
→We inserted “cloning and functional expression of the equine luteinizing hormone/ chorionic gonadotropin receptor” Saint-Dizier et al., 2004 (Journal of endocrinology). In the reference 15.
- The authors should mount the study better by reframing the abstract. Instead of detailing the results, they should specify the broader implications of this study and how this knowledge will benefit the scientific community.
→We inserted and changed “Specifically, the N174Q and N195Q mutants exhibited markedly reduced expression levels, reaching approximately 15.3% and 2.5%, respectively, of that observed for wild-type equine LH/CGR. Correspondingly, the cAMP ECâ‚…â‚€ values were decreased by 7.6-fold and 5.6-fold, respectively. Notably, the N195Q mutant displayed an almost complete loss of cAMP activity, even at high concentrations of recombinant eCG, suggesting a critical role for this glycosylation site in receptor function. Despite these alterations, Western blot analysis revealed that pERK1/2 phosphorylation peaked at 5 minutes following agonist stimulation across all mutants, indicating that the ERK1/2 signaling pathway remains functionally intact.” In the abstract section.
- The authors dedicate significant portion of the manuscript to a single experiment measuring cAMP levels in presence of the LH/CGR mutants. They present the data as individual graphs, in a table and then in a bar plot (Figure 3, 4 and Table 1). I suggest that the authors only stick to the bar plot and move the graphs and table to the SI.
→In this study, the cAMP results were considered the most important in evaluating the functional role of the glycosylation sites. Therefore, these data were first presented in Figure 3 in comparison with the wild type. The basal levels, EC50, and Rmax values were then further summarized and presented in Table 1 and Figure 4. Since the authors believe this is the most critical part of the study, the results were intentionally separated into both a figure and a table for clarity. Accordingly, if the reviewer is willing to allow it, the authors would prefer that these results be included in the main text rather than in the supplementary materials.
- The same point also applies to Figure 6, 7 and Table 2. In this case, Table 2 seems to represent the most relevant information. Please move the figures to SI.
→As this part is also as important as the cAMP results, we have included it together for clarity and emphasis. Therefore, we kindly ask for the reviewer’s generous consideration in allowing it to remain in the main text.
- Can the authors discuss why the trend in N195Q is completely different compared to the other mutants?
→We inserted “The N195Q mutant shows a distinctly different result compared to other glycosylation site mutants. This unique outcome may be attributed to the significantly reduced receptor expression observed with the N195Q mutation, which is likely closely related to improper receptor folding. As a result, it can be concluded that there is minimal loss at the cell surface in this mutant” in the discussion section in line 511-515.
- The major concern in this work is that the authors do not verify the presence of the N-glycosylation PTM, and whether it is affected by introducing the mutants. The authors should carry out an orthogonal measurement, such as mass spectrometry to determine the extent of N-glycosylation. Alternatively, they should cite literature showing that the side-directed mutagenesis does not affect N-glycosylation at the other sites.
→The authors substituted asparagine (N) with glutamine (Q) at the N-linked glycosylation sites, as Q is known to have the least impact on post-translational modifications (PTMs). Therefore, the N195Q mutation was designed specifically to disrupt glycosylation. As discussed in the Discussion section, this substitution appears to have affected PTMs, resulting in minimal receptor expression at the cell surface. Although orthogonal measurements such as mass spectrometry, as suggested by the reviewer, were not performed, we believe that the results can still be reasonably interpreted based on established methods and previous studies.
- Line 361 has the first and only mention of N201Q. I assume this is a typo and should actually be N195Q.
→We changed “N201Q” to “N195Q” by reviewer comment in the line 383.
Reviewer 3 Report
Comments and Suggestions for Authors
The work by Byambaragchaa et al. characterizes the function of each N-glycosylation site of the extracellular domain of the LH/CGR by means of single point mutations. The outcomes at cellular level have been analyzed by measuring the level of cell surface protein expression, cAMP signaling, and pERK1/2 pathway.
Although this reviewer considers the study of protein site-specific glycosylation and its impact on cellular outcomes extremely relevant, this study presents several criticism that require extensive revision.
Minor revision:
-Abbreviations in the abstract should be avoided, these includes: “cAMP activation” line 21; “pERK1/2” line 22.
-line 27: “The N195Q mutant showed a significantly prolonged delay.” This sentence is not clear. Please revise and clarify what prolonged delay means in this context.
-line 205. “Specifically, an additional glycosylation site was uniquely present at position 50 in equine LH/CGR.” Additional with respect to?
Line 207. “The N201 site, located in exon 3 of LH/CGR, and the other three sites (N291, N299, and N313), located in exon 10 of LH/CGR, were identified by amino acid comparison analysis.” What the authors mean with “amino acid comparison analysis” what are they comparing and with what?
-Line 282: “N199Q” should be N195Q.
Major revision
-In line 80 of the introduction. The authors discuss, based on previous study, that “in glycoprotein hormone receptors, N-linked glycosylation sites do not directly participate in receptor binding but are known to play a key role in selective cell surface expression and cAMP activation.” This reviewer find such a conclusion an overstatement since others role of glycosylation are not considered or discussed. Those include the role of glycosylation in protein stability and turnover and/or the role of glycans in mediate interactions with others proteins or receptors, such as lectins, that can contribute to regulate, or even block, hormone binding.
- In the results section, the authors discuss the analysis of cell surface expression of the LH/CGR WT and generated mutants. In line 221, the authors claim that “Therefore, it is speculated that the glycosylation site mutations at N174 and N195, located in exons 6 and 7, respectively, have a particularly significant impact on cell surface expression level.” Why is this a speculation? The expression level at cellular surface has been experimentally determined by ELISA, thus I do not see a speculation, but a demonstration.
-Figure 2. What the single asterisk stay for?
-Regarding the ELISA experiments. Why the authors performed ELISA experiments only for the cell surface LH/CGR? If they perform the same experiments, or western blot, with the cell lysate, this can provide essential information about the impact of site-specific glycosylation on total protein expression level over the protein transport to the cell surface. In fact, it is very well known that N-glycans at specific glycosylation sites serve as protein folding quality control, doi: 10.1101/glycobiology.4e.39. Such an experiments would clarify the role of each glycosylation site in the protein.
-ELISA experiments are very sensitive. It would be useful to measure cell surface levels by a complementary/quantitative technique such as flow cytometry.
-Figure 3. The authors report the total cAMP levels for each mutant. Are those values normalized with respect to the total amount f cell surface LH/CGR? If not, how meaningful is such analysis?
-line 252. How do authors explain N291Q? I mean, similar EC to wt but significantly lower R?
-When the authors evaluate the cAMP levels in the presence of the ligand for the different mutants. Given the difference observed between the magnitude of the LH/CGR expression level and the magnitude of cAMP level, they conclude that there has to be a functional difference between the different mutants. This would be true if the changes in cAMP levels are significantly lower than the protein expression level. I mean, the protein is expressed but not functional. However, for the N174Q mutant, the protein expression level drops 6.6 fold and the cAMP level 9.6 fold, in case of N195Q the protein expression level drop 40 fold and the cAMP level 5.6 fold.
-To demonstrate that the different mutants are responsible for a different pERK, the authors measure the amount of pERK by WB. It would be useful to measure the level of others proteins in the pERK pathway to ensure that this is the cell response pathway affected by the LH/CGR mutants.
-In that sense, the authors demonstrate changes in pERK levels, but unexpectedly, they also observe changes in ERK total (figure 8). While it is reasonable that the level of pERK changes as function of time (5 min), it is very difficult to justify that the ERK total also change. Including loading control, such as actin or tubulin, is required.
Author Response
Reviewer 3
The work by Byambaragchaa et al. characterizes the function of each N-glycosylation site of the extracellular domain of the LH/CGR by means of single point mutations. The outcomes at cellular level have been analyzed by measuring the level of cell surface protein expression, cAMP signaling, and pERK1/2 pathway.
Although this reviewer considers the study of protein site-specific glycosylation and its impact on cellular outcomes extremely relevant, this study presents several criticism that require extensive revision.
Minor revision:
-Abbreviations in the abstract should be avoided, these includes: “cAMP activation” line 21; “pERK1/2” line 22.
→We changed the abbreviation in the abstract to full name et al., cAMP and pERK1/2.
-line 27: “The N195Q mutant showed a significantly prolonged delay.” This sentence is not clear. Please revise and clarify what prolonged delay means in this context.
→ “The N195Q mutant showed a significantly prolonged delay” was changed to “The N195Q mutant was nearly completely impaired, even at high rec-eCG concentrations” in the line 27-28.
-line 205. “Specifically, an additional glycosylation site was uniquely present at position 50 in equine LH/CGR.” Additional with respect to?
→ We inserted “which is absent in the LH/CGR of other mammalian species” after “Specifically, an additional glycosylation site was uniquely present at position 50 in equine LH/CGR” in the line 212-213.
Line 207. “The N201 site, located in exon 3 of LH/CGR, and the other three sites (N291, N299, and N313), located in exon 10 of LH/CGR, were identified by amino acid comparison analysis.” What the authors mean with “amino acid comparison analysis” what are they comparing and with what?
→ This explains the exon positions corresponding to each glycosylation site, identified through comparative amino acid analysis of mammalian LH/CGRs based on whole-genome sequence alignment
-Line 282: “N199Q” should be N195Q.
→ We changed “N199” to “N195”.
Major revision
-In line 80 of the introduction. The authors discuss, based on previous study, that “in glycoprotein hormone receptors, N-linked glycosylation sites do not directly participate in receptor binding but are known to play a key role in selective cell surface expression and cAMP activation.” This reviewer find such a conclusion an overstatement since others role of glycosylation are not considered or discussed. Those include the role of glycosylation in protein stability and turnover and/or the role of glycans in mediate interactions with others proteins or receptors, such as lectins, that can contribute to regulate, or even block, hormone binding.
→ We inserted “Although the receptor binding analysis was not performed in this study, the N195Q mutant was found to have the greatest impact on cAMP activity. Therefore, it is suggested that the glycosylation sites play important roles in receptor stability, turnover and interactions with other proteins” in the discussion section of the line 510-513.
- In the results section, the authors discuss the analysis of cell surface expression of the LH/CGR WT and generated mutants. In line 221, the authors claim that “Therefore, it is speculated that the glycosylation site mutations at N174 and N195, located in exons 6 and 7, respectively, have a particularly significant impact on cell surface expression level.” Why is this a speculation? The expression level at cellular surface has been experimentally determined by ELISA, thus I do not see a speculation, but a demonstration.
→ We changed “ speculated” to “suggested”
-Figure 2. What the single asterisk stay for?
→ The last line of the legend in Figure 2 was corrected by replacing the two asterisks with a single asterisk.
-Regarding the ELISA experiments. Why the authors performed ELISA experiments only for the cell surface LH/CGR? If they perform the same experiments, or western blot, with the cell lysate, this can provide essential information about the impact of site-specific glycosylation on total protein expression level over the protein transport to the cell surface. In fact, it is very well known that N-glycans at specific glycosylation sites serve as protein folding quality control, doi: 10.1101/glycobiology.4e.39. Such an experiments would clarify the role of each glycosylation site in the protein.
-ELISA experiments are very sensitive. It would be useful to measure cell surface levels by a complementary/quantitative technique such as flow cytometry.
→ Of course, as the reviewer pointed out, Western blotting is one possible analysis method. However, we chose to use ELISA instead, as it is more sensitive for detecting cell surface expression. Additionally, ELISA allows for the assessment of cell surface loss, which was also part of our analysis. Therefore, this method was selected.
-Figure 3. The authors report the total cAMP levels for each mutant. Are those values normalized with respect to the total amount of cell surface LH/CGR? If not, how meaningful is such analysis?
→ As described in the Methods section, equal amounts of plasmid vectors encoding each glycosylation site mutant were transiently transfected into the cells to ensure consistent expression conditions. Following transfection, the total amount of cAMP produced in the cells was measured as an indicator of receptor activity. Since the primary goal was to assess intracellular signaling rather than receptor localization, the analysis did not take into account the differences in cell surface receptor expression levels.
-line 252. How do authors explain N291Q? I mean, similar EC to wt but significantly lower R?
→The EC50 and Rmax values for cAMP production by each mutant were directly derived from the analysis performed using GraphPad, a software program for cAMP response analysis. Therefore, the results were reported as calculated by the program. In fact, the N291Q mutant showed a lower Rmax value compared to the wild type, with a 0.72-fold decrease.
-When the authors evaluate the cAMP levels in the presence of the ligand for the different mutants. Given the difference observed between the magnitude of the LH/CGR expression level and the magnitude of cAMP level, they conclude that there has to be a functional difference between the different mutants. This would be true if the changes in cAMP levels are significantly lower than the protein expression level. I mean, the protein is expressed but not functional. However, for the N174Q mutant, the protein expression level drops 6.6 fold and the cAMP level 9.6 fold, in case of N195Q the protein expression level drop 40 fold and the cAMP level 5.6 fold.
→The correlation between cell surface expression levels and cAMP production was a matter of serious consideration by the authors of this study. In the case of the N50Q mutant, for example, although its expression level was significantly reduced, the EC50 value for cAMP production showed a slight increase. Therefore, it is considered that cell surface expression levels do not necessarily correspond directly to the amount of cAMP generated.
-To demonstrate that the different mutants are responsible for a different pERK, the authors measure the amount of pERK by WB. It would be useful to measure the level of others proteins in the pERK pathway to ensure that this is the cell response pathway affected by the LH/CGR mutants.
→At this stage, the primary focus of this study was on investigating the relationship between glycosylation sites and PKA signaling, specifically in terms of their effects on cAMP and pERK1/2. Therefore, receptor-level factors influenced by LH/CGR were not taken into account. Further investigation in this area is planned for future studies.
-In that sense, the authors demonstrate changes in pERK levels, but unexpectedly, they also observe changes in ERK total (figure 8). While it is reasonable that the level of pERK changes as function of time (5 min), it is very difficult to justify that the ERK total also change. Including loading control, such as actin or tubulin, is required.
→To date, our laboratory has published numerous studies on pERK1/2, and we have consistently used SuperSignal West Femto Maximum for detecting total ERK. This likely resulted in the detection of a strong signal. However, this did not pose a significant issue for quantitative analysis.
Round 2
Reviewer 1 Report
Comments and Suggestions for Authors
Accept
Author Response
Reviewer’s comment
We could not have any revised by accepted.

Reviewer 3 Report
Comments and Suggestions for Authors
Major revision
-Regarding the ELISA experiments. Why the authors performed ELISA experiments only for the cell surface LH/CGR? If they perform the same experiments, or western blot, with the cell lysate, this can provide essential information about the impact of site-specific glycosylation on total protein expression level over the protein transport to the cell surface. In fact, it is very well known that N-glycans at specific glycosylation sites serve as protein folding quality control, doi: 10.1101/glycobiology.4e.39. Such an experiments would clarify the role of each glycosylation site in the protein.
-ELISA experiments are very sensitive. It would be useful to measure cell surface levels by a complementary/quantitative technique such as flow cytometry.
→ Of course, as the reviewer pointed out, Western blotting is one possible analysis method. However, we chose to use ELISA instead, as it is more sensitive for detecting cell surface expression. Additionally, ELISA allows for the assessment of cell surface loss, which was also part of our analysis. Therefore, this method was selected.
The authors did not address the point raised by the reviewer on the estimation of the amount of LH/CGR on the cell surface vs total expression for each of the generated mutants. This is important since the authors themselves comment that “cell surface expression levels do not necessarily correspond directly to the amount of cAMP generated” and that “the primary goal was to assess intracellular signaling rather than receptor localization, the analysis did not take into account the differences in cell surface receptor expression levels.”
-Figure 3. The authors report the total cAMP levels for each mutant. Are those values normalized with respect to the total amount of cell surface LH/CGR? If not, how meaningful is such analysis?
→ As described in the Methods section, equal amounts of plasmid vectors encoding each glycosylation site mutant were transiently transfected into the cells to ensure consistent expression conditions. Following transfection, the total amount of cAMP produced in the cells was measured as an indicator of receptor activity. Since the primary goal was to assess intracellular signaling rather than receptor localization, the analysis did not take into account the differences in cell surface receptor expression levels.
The use of an equal amounts of plasmid vectors does not necessary translate into consistent expression levels. Instead, if the mutation of a given glycosylation site leads to a not functional/stable protein this is quickly degraded. Thus, the calculated amount of cAMP should be normalized with respect to the amount of functional protein.
-line 252. How do authors explain N291Q? I mean, similar EC to wt but significantly lower R?
→The EC50 and Rmax values for cAMP production by each mutant were directly derived from the analysis performed using GraphPad, a software program for cAMP response analysis. Therefore, the results were reported as calculated by the program. In fact, the N291Q mutant showed a lower Rmax value compared to the wild type, with a 0.72-fold decrease.
The way how the EC50 and Rmax values have been derived was already clear. What is not clear is interpretation of the results.
-When the authors evaluate the cAMP levels in the presence of the ligand for the different mutants. Given the difference observed between the magnitude of the LH/CGR expression level and the magnitude of cAMP level, they conclude that there has to be a functional difference between the different mutants. This would be true if the changes in cAMP levels are significantly lower than the protein expression level. I mean, the protein is expressed but not functional. However, for the N174Q mutant, the protein expression level drops 6.6 fold and the cAMP level 9.6 fold, in case of N195Q the protein expression level drop 40 fold and the cAMP level 5.6 fold.
→The correlation between cell surface expression levels and cAMP production was a matter of serious consideration by the authors of this study. In the case of the N50Q mutant, for example, although its expression level was significantly reduced, the EC50 value for cAMP production showed a slight increase. Therefore, it is considered that cell surface expression levels do not necessarily correspond directly to the amount of cAMP generated.
Thus, how those results must be interpreted?
-To demonstrate that the different mutants are responsible for a different pERK, the authors measure the amount of pERK by WB. It would be useful to measure the level of others proteins in the pERK pathway to ensure that this is the cell response pathway affected by the LH/CGR mutants.
→At this stage, the primary focus of this study was on investigating the relationship between glycosylation sites and PKA signaling, specifically in terms of their effects on cAMP and pERK1/2. Therefore, receptor-level factors influenced by LH/CGR were not taken into account. Further investigation in this area is planned for future studies.
I do not understand the authors reply. I suggested to quantify the level of other proteins in the pERK pathway in order to corroborate that this is the cellular pathway affected by the different LH/CGR mutants.
-In that sense, the authors demonstrate changes in pERK levels, but unexpectedly, they also observe changes in ERK total (figure 8). While it is reasonable that the level of pERK changes as function of time (5 min), it is very difficult to justify that the ERK total also change. Including loading control, such as actin or tubulin, is required.
→To date, our laboratory has published numerous studies on pERK1/2, and we have consistently used SuperSignal West Femto Maximum for detecting total ERK. This likely resulted in the detection of a strong signal. However, this did not pose a significant issue for quantitative analysis.
The request to include a loading control is a must.
In general, the authors did not make any effort to address the points raised by the reviewer. Thus, I will be more clear. In the conclusion section:
“Taken together, this study demonstrated that the N195Q mutants had the lowest expression on the cell surface, followed by a basal cAMP response, which was not completely impaired. [….] Thus, the potential N195 glycosylation site located in exon 7 of LH/CGR may serve as a useful model for impairing cAMP signaling and downregulating pathways in receptor-mediated complexes.” But, the authors discuss that “cell surface expression levels do not necessarily correspond directly to the amount of cAMP generated” thus, I do not see a meaningful conclusion.
“The lack of glycosylation at this site, along with its substitution with Gln, is presumed to have induced a conformational change after PTMs”. This can’t be a conclusion since the effect of N195Q on protein structure and conformation has not been investigated in this work.
“Thus, the potential N195 glycosylation site located in exon 7 of LH/CGR may serve as a useful model for impairing cAMP signaling and downregulating pathways in receptor-mediated complexes.” How? Please expand.
Author Response
Major revision
-Regarding the ELISA experiments. Why the authors performed ELISA experiments only for the cell surface LH/CGR? If they perform the same experiments, or western blot, with the cell lysate, this can provide essential information about the impact of site-specific glycosylation on total protein expression level over the protein transport to the cell surface. In fact, it is very well known that N-glycans at specific glycosylation sites serve as protein folding quality control, doi: 10.1101/glycobiology.4e.39. Such an experiments would clarify the role of each glycosylation site in the protein.
-ELISA experiments are very sensitive. It would be useful to measure cell surface levels by a complementary/quantitative technique such as flow cytometry.
→ Of course, as the reviewer pointed out, Western blotting is one possible analysis method. However, we chose to use ELISA instead, as it is more sensitive for detecting cell surface expression. Additionally, ELISA allows for the assessment of cell surface loss, which was also part of our analysis. Therefore, this method was selected.
The authors did not address the point raised by the reviewer on the estimation of the amount of LH/CGR on the cell surface vs total expression for each of the generated mutants. This is important since the authors themselves comment that “cell surface expression levels do not necessarily correspond directly to the amount of cAMP generated” and that “the primary goal was to assess intracellular signaling rather than receptor localization, the analysis did not take into account the differences in cell surface receptor expression levels.”
→ In this study, we emphasized in three parts of the Discussion that receptor expression is essential for the N174Q and N195Q mutants (lines 491–495), and that specific N-linked glycosylation sites, rather than all glycosylation sites, are selectively associated with cAMP activity and cell surface loss (lines 520–523). In particular, we highlighted that the N195Q mutant plays a crucial role in receptor expression, cell surface loss, and PKA activation. Therefore, we did not state, as the reviewer suggested, that the level of cell surface expression is not essential for cAMP production.
We have also thoroughly revised this part to make it clear in Line 482-485. Therefore, not all the N-linked glycosylation sites of equine LH/CGR play a crucial role; rather, specific sites are selectively involved in cell surface expression, receptor loss on the cell surface, and PKA-mediated physiological activity.
Changed to “These findings indicate that not all N-linked glycosylation sites of equine LH/CGR are essential; instead, distinct glycosylation sites selectively contribute to specific receptor functions, such as proper trafficking to the cell surface, regulation of receptor internalization, and modulation of PKA-mediated physiological responses”/
-Figure 3. The authors report the total cAMP levels for each mutant. Are those values normalized with respect to the total amount of cell surface LH/CGR? If not, how meaningful is such analysis?
→ As described in the Methods section, equal amounts of plasmid vectors encoding each glycosylation site mutant were transiently transfected into the cells to ensure consistent expression conditions. Following transfection, the total amount of cAMP produced in the cells was measured as an indicator of receptor activity. Since the primary goal was to assess intracellular signaling rather than receptor localization, the analysis did not take into account the differences in cell surface receptor expression levels.
The use of an equal amounts of plasmid vectors does not necessary translate into consistent expression levels. Instead, if the mutation of a given glycosylation site leads to a not functional/stable protein this is quickly degraded. Thus, the calculated amount of cAMP should be normalized with respect to the amount of functional protein.
→ As presented in this study, the purpose was to analyze the functional role of each glycosylation mutant. According to the reviewer's suggestion, the appropriate approach would involve generating stable cell lines and selecting clones with expression levels comparable to the wild-type, which could then be used for experimentation. However, in this study, we employed a transient expression system and analyzed cAMP levels in each transfected cell population, which allowed us to determine which specific mutant had the greatest impact on cAMP production. Therefore, since we used equal amounts of plasmid for transient transfection and assessed cAMP production under these conditions, we find it inappropriate to normalize the data based on functional protein levels. We kindly ask for your understanding on this point.
-line 252. How do authors explain N291Q? I mean, similar EC to wt but significantly lower R?
→The EC50 and Rmax values for cAMP production by each mutant were directly derived from the analysis performed using GraphPad, a software program for cAMP response analysis. Therefore, the results were reported as calculated by the program. In fact, the N291Q mutant showed a lower Rmax value compared to the wild type, with a 0.72-fold decrease.
The way how the EC50 and Rmax values have been derived was already clear. What is not clear is interpretation of the results.
→ we changed “increased’ to “decreased” in the Line 267.
-When the authors evaluate the cAMP levels in the presence of the ligand for the different mutants. Given the difference observed between the magnitude of the LH/CGR expression level and the magnitude of cAMP level, they conclude that there has to be a functional difference between the different mutants. This would be true if the changes in cAMP levels are significantly lower than the protein expression level. I mean, the protein is expressed but not functional. However, for the N174Q mutant, the protein expression level drops 6.6 fold and the cAMP level 9.6 fold, in case of N195Q the protein expression level drop 40 fold and the cAMP level 5.6 fold.
→The correlation between cell surface expression levels and cAMP production was a matter of serious consideration by the authors of this study. In the case of the N50Q mutant, for example, although its expression level was significantly reduced, the EC50 value for cAMP production showed a slight increase. Therefore, it is considered that cell surface expression levels do not necessarily correspond directly to the amount of cAMP generated.
Thus, how those results must be interpreted?
→ we inserted in the Line 474-482 “The N50Q mutant, located in exon 3—which is not present in other mammalian receptors—exhibited distinct outcomes in both expression levels and cAMP production. Although the cell surface expression of the N50Q mutant was significantly reduced by approximately 38%, its cAMP production was paradoxically increased. This unique observation, specific to this mutant, suggests that there may not be a direct correlation between cell surface expression and cAMP production. One possible explanation is that this region lies at the very beginning of the exon, which may have limited impact on post-translational modifications (PTMs), while promoting a folding structure that favors PKA activation.
-To demonstrate that the different mutants are responsible for a different pERK, the authors measure the amount of pERK by WB. It would be useful to measure the level of others proteins in the pERK pathway to ensure that this is the cell response pathway affected by the LH/CGR mutants.
→At this stage, the primary focus of this study was on investigating the relationship between glycosylation sites and PKA signaling, specifically in terms of their effects on cAMP and pERK1/2. Therefore, receptor-level factors influenced by LH/CGR were not taken into account. Further investigation in this area is planned for future studies.
I do not understand the authors reply. I suggested to quantify the level of other proteins in the pERK pathway in order to corroborate that this is the cellular pathway affected by the different LH/CGR mutants.
→ At this stage, we have not conducted analyses on other proteins. Currently, we are performing additional experiments using β-arrestin 1 and 2 knockout (KO) cells to investigate this aspect further, and therefore, it may take some additional time.
-In that sense, the authors demonstrate changes in pERK levels, but unexpectedly, they also observe changes in ERK total (figure 8). While it is reasonable that the level of pERK changes as function of time (5 min), it is very difficult to justify that the ERK total also change. Including loading control, such as actin or tubulin, is required.
→To date, our laboratory has published numerous studies on pERK1/2, and we have consistently used SuperSignal West Femto Maximum for detecting total ERK. This likely resulted in the detection of a strong signal. However, this did not pose a significant issue for quantitative analysis.
The request to include a loading control is a must.
→ Yes, according to the reviwer’s comment, we are going to prepare the internal control protein.
Thus, we need to take a time for this experiment.
In general, the authors did not make any effort to address the points raised by the reviewer. Thus, I will be more clear. In the conclusion section:
“Taken together, this study demonstrated that the N195Q mutants had the lowest expression on the cell surface, followed by a basal cAMP response, which was not completely impaired. [….] Thus, the potential N195 glycosylation site located in exon 7 of LH/CGR may serve as a useful model for impairing cAMP signaling and downregulating pathways in receptor-mediated complexes.” But, the authors discuss that “cell surface expression levels do not necessarily correspond directly to the amount of cAMP generated” thus, I do not see a meaningful conclusion.
→ We revised in the Line 491-494 “These findings indicate that not all N-linked glycosylation sites of equine LH/CGR are essential; instead, distinct glycosylation sites selectively contribute to specific receptor functions, such as proper trafficking to the cell surface, regulation of receptor internalization, and modulation of PKA-mediated physiological responses”.
“The lack of glycosylation at this site, along with its substitution with Gln, is presumed to have induced a conformational change after PTMs”. This can’t be a conclusion since the effect of N195Q on protein structure and conformation has not been investigated in this work.
→ we revised “Although this study did not directly examine structural changes at the molecular level, it is presumed that the absence of glycosylation at this specific site, combined with the substitution of asparagine with glutamine, may have led to conformational alterations in the receptor structure following post-translational modifications (PTMs).”
“Thus, the potential N195 glycosylation site located in exon 7 of LH/CGR may serve as a useful model for impairing cAMP signaling and downregulating pathways in receptor-mediated complexes.”
How? Please expand.
→ These structural shifts could, in turn, influence receptor folding, trafficking, and overall functionality. In this context, the potential N195 glycosylation site, located within exon 7 of the LH/CGR, may represent a valuable model system for exploring the mechanistic roles of site-specific glycosylation in modulating receptor-mediated signaling. Specifically, it could provide important insights into how the disruption of glycosylation can impair cAMP signaling efficiency and alter downstream regulatory pathways involved in GPCR function.
